# Independent of physical activity, volumetric muscle loss injury in a murine model impairs whole-body metabolism

Kyle A. Dalske[1][ᵒ], Christiana J. Raymond-Pope[1][ᵒ], Jennifer McFaline-Figueroa[2,3], Alec M. Basten[1], Jarrod A. Call[2,3], Sarah M. Greising[1]*

1 School of Kinesiology, University of Minnesota, Minneapolis, MN, United States of America, 2 Department of Kinesiology, University of Georgia, Athens, GA, United States of America, 3 Regenerative Bioscience Center, University of Georgia, Athens, GA, United States of America

ᵒ These authors contributed equally to this work.
* grei0064@umn.edu

## Abstract

Volumetric muscle loss (VML) injuries result in a non-recoverable loss of muscle tissue and function due to trauma or surgery. Reductions in physical activity increase the risk of metabolic comorbidities over time, and it is likely that VML may reduce whole-body activity. However, these aspects remain uncharacterized following injury. Our goal was to characterize the impact of VML on whole-body physical activity and metabolism, and to further investigate possible muscle-specific metabolic changes. Adult male C57Bl/6J (n = 28) mice underwent a standardized VML injury to the posterior compartment of the hind limb, or served as injury naïve controls. Mice underwent longitudinal evaluation of whole-body physical activity and metabolism in specialized cages up to three times over the course of 8 weeks. At terminal time points of 4- and 8-weeks post-VML *in vivo* muscle function of the posterior compartment was evaluated. Additionally, the gastrocnemius muscle was collected to understand histological and biochemical changes in the muscle remaining after VML. The VML injury did not alter the physical activity of mice. However, there was a noted reduction in whole-body metabolism and diurnal fluctuations between lipid and carbohydrate oxidation were also reduced, largely driven by lower carbohydrate utilization during active hours. Following VML, muscle-specific changes indicate a decreased proportion of fast (i.e., type IIb and IIx) and a greater proportion of slow (i.e., type I and IIa) fibers. However, there were minimal changes in the capillarity and metabolic biochemical activity properties of the gastrocnemius muscle, suggesting a miss-match in capacity to support the physiologic needs of the fibers. These novel findings indicate that following VML, independent of changes in physical activity, there is whole-body diurnal metabolic inflexibility. Supporting future investigations into the chronic and overlooked co-morbidities of VML injury.

**Data Availability Statement:** All relevant data are within the paper.

**Funding:** Funding through the Congressionally Directed Medical Research Program, Clinical &

Rehabilitative Medicine Research Program: W81XWH-18-1-0710 (JAC and SMG) and the National Institutes of Health T32-AR050938 (CJRP). The funders had no role in study design, data collection and analysis, decision to publish, or preparation of the manuscript.

Competing interests: The authors have declared that no competing interests exist.

## Introduction

Diurnal alterations in metabolic fuel selection, i.e., carbohydrates and lipids reflect the body's capacity to maintain energy homeostasis in response to daily fluctuations in activity and substrate availability (e.g., after a meal). Also known as metabolic flexibility [1], the changing reliance on carbohydrate and/or lipid oxidation is influenced by a number of tissues and hormones (e.g., insulin and glucagon) as well as molecular signaling cascades that regulate substrate availability at the mitochondria (e.g., malonyl-CoA inhibition of CPT-1-induced free fatty acid uptake) [2]. Metabolically healthy individuals are able to efficiently transition between fuel sources, such as having a greater carbohydrate oxidation rate immediately following a meal and a greater lipid oxidation rate during times of fasting to help preserve blood glucose levels [3]. Energetic demands during exercise also require transitions between states of predominately lipid (i.e., low-intensity exercise) and carbohydrate oxidation (i.e., high-intensity) as the body matches fuel availability to demand and maintains normoglycemia. In contrast, individuals with pathologic conditions of type II diabetes and obesity have greater prevalence of metabolic inflexibility, or an inability to transition between fuel sources [4]. In addition to disease-related metabolic inflexibility, sedentary behavior is linked to greater all-cause mortality, atherosclerosis, reduced cardiac function, type II diabetes, and obesity [5], and physical inactivity has been implicated as a factor driving the onset of metabolic inflexibility [6,7].

As the primary organ of movement and thermogenesis [8], skeletal muscle must utilize fuel from energy-yielding sources, primarily carbohydrates for high workloads [9] and lipids for low workloads and rest [10]. Various conditions may impair the ability of muscle to do physical work, increasing risk of metabolic impairment both acutely and chronically. Large-scale orthopaedic traumas have broad implications on long-term health and function and are an example of a condition that could potentially increase risk for metabolic comorbidities. For example, in patients who sustained fracture to the upper or lower extremities, there are known decreases in physical activity and increases in sedentarism acutely following injury [11,12]. Similarly, various orthopaedic injuries and traumas result in decreases in physical activity as early as three months after initial injury [13]. One injury of particular significance is volumetric muscle loss (VML), in which a substantial volume of a muscle or muscle unit is abruptly removed, resulting in chronic loss of muscle function and range of motion [14]. Currently, the long-term comorbidities secondary to VML injury are not well understood, however, it is expected that there is a loss of skeletal muscle metabolic and physical activity. It is anticipated that after initial VML injury there will be periods of reduced mobilization that are unavoidable during acute and prolonged care. It is thus hypothesized, that patients with VML injury could be more likely to undergo periods of reduced physical activity and subsequently develop chronic metabolic inflexibility.

We posit that the sequela of VML injury includes impaired whole-body metabolism due to low physical activity and the loss of skeletal muscle tissue. Previous work in animal models of VML has indicated that skeletal muscle specific oxidative function is impaired following injury [15,16], and that the muscle remaining after VML injury has reduced metabolic plasticity compared to healthy, uninjured muscle in response to rehabilitation [17]. Dysfunction of skeletal muscle metabolism may impede the endogenous healing process and induce further chronic dysfunction. To date it is unknown if VML injury impacts whole-body metabolism and physical activity both in animal models of VML injury or the patient population. The studies herein were designed to first understand if VML injury results in reduced physical activity and whole-body metabolism, corresponding to chronic metabolic inflexibility. Second, to evaluate contractile and oxidative physiology of the skeletal muscle fibers after VML injury.

## Materials and methods

### Animals care and study design

Adult male C57Bl/6J (n = 28) mice were purchased from Jackson Laboratories (Stock # 000664; Bar Harber, ME). Mice were allowed at least one week of acclimation to the facility prior to initiating any aspect of the study protocol. All protocols and animal care guidelines were approved by the University of Minnesota Institutional Animal Care and Use Committee (Protocol 1803-35671A), in compliance with the National Institute of Health Guidelines and the United States Department of Agriculture. All studies were conducted in compliance with the Animal Welfare Act, the Implementing Animal Welfare Regulations and in accordance with the principles of the Guide for the Care and Use of Laboratory Animals.

Mice were housed on a 12-hour light-dark cycle (light phase begins at 06:00) with *ad libitum* access to standard chow and water. At ~12.5 weeks of age mice were allocated by rank order scheme assignment of respiratory exchange ratio, into VML injury (n = 12), sham surgery (n = 12), or injury naïve controls (n = 4). Mice in the sham and naïve groups were not different and combined into a control group. Physical activity and metabolic assessments were conducted using Comprehensive Lab Animal Monitoring System (CLAMS; Columbus Instruments) at approximately 1-week pre-surgery and 2- and 6-weeks post-surgery for all animals. *In vivo* muscle function was assessed terminally 4- and 8-weeks post-surgery. Immediately following muscle function assessments, mice were euthanized with pentobarbital (>100mg/kg; s. q.), and skeletal muscles were harvested and frozen at -80°C for later analyses.

### Physical activity and metabolic assessment

Mice were acclimated to individual CLAMS cages for 24-hours prior to each data collection period. The CLAMS system is a closed chamber that measures ventilation gases $O_2$ and $CO_2$ normalized to body mass (ml/kg/hr) to quantify metabolic activity. The respiratory exchange ratio (RER) is calculated as the volume ratio of $CO_2$ and $O_2$. Calorific value (CV) is the relationship between heat and oxygen consumption, as described in Graham Lusk's "*The Elements of the Science of Nutrition*" [18] by the following equation: $CV = 3.815 + 1.232 * RER$. Further, energy expenditure (i.e., heat) is calculated as the product of the CV and the $VO_2$ of the mouse and represents the metabolic rate. Carbohydrate and lipid oxidation rates (g/min) were calculated using the following equations: carbohydrate oxidation = $(VCO_2 * 4.585)–(VO_2 * 3.226)$; lipid oxidation = $(VO_2 * 1.695)–(VCO_2 * 1.701)$.

Each CLAMS cage is equipped with infrared beams in the X, Y, and Z axes that register physical movement when the beam array is broken. Individual beam breaks are classified as activity counts. When the mouse is moving in the same position (i.e., grooming, drinking, and feeding), the movement is tracked as a count. As multiple beams in sequence are broken, the movement is tracked as ambulatory counts. Beams along the X and Y axis are spaced at 1.27cm intervals, allowing quantification of ambulatory distance.

Data were collected and analyzed using CLAMS data examination Tool (Clax, v2.2.15; Columbus Instruments), a software integrated with CLAMS for the 24-hour periods. Data for total counts and ambulatory counts were collected in 10-second increments, and RER, metabolic rate, $VO_2$ and $VCO_2$ were collected in 4-minute increments. All data were subsequently exported for later analyses.

Overall data were analyzed in bins of 24, 12 (active vs. inactive time periods), 6, and 1 hour and presented as a mean over the time period. Metabolic rate flux and metabolic flexibility were measured by assessing the range of metabolic rate and RER, respectively, as the delta between 12-hour active and inactive values. Each incremental RER and metabolic rate data

were also used to calculate hourly moving averages using an average of every 15, 4-minute increment. The moving averages of metabolic rate and RER were used to evaluate the area under the curve (AUC) as a more specific measure of metabolic rate flux and metabolic flexibility, respectively.

## Volumetric muscle loss (VML) surgical procedure

As previously described [16,17,19], a full thickness VML injury was surgically created to the posterior compartment muscle group (gastrocnemius, soleus, and plantaris muscles). A subset of mice underwent sham VML procedures or no surgical procedure (injury naïve). About 2 hours prior to surgery mice received SR buprenorphine (1.2mg/kg; s.q.) for pain management. Mice were anesthetized using isoflurane (~2.0%) under aseptic surgical conditions. A posterior-lateral incision was made through the skin to reveal the gastrocnemius muscle. Blunt dissection was used to isolate the posterior muscle compartment, and a metal plate was inserted between the tibia and the deep aspect of the soleus. A 4 mm punch biopsy (20.9±4.9 mg, ~15% volume loss of muscle) was performed on the middle third of the muscle compartment. Any bleeding was stopped with pressure. Incisions were closed with 6.0 silk sutures and animals were monitored through recovery.

## *In vivo* muscle function

At the terminal time point (4- or 8-weeks post-surgical) the isometric torque of the posterior compartment muscle was evaluated as previously described [16,17]. Briefly, *in vivo* isometric torque was measured in anesthetized mice (isoflurane 1.5–2.0%) while body temperature was maintained at 37˚sC. The foot was positioned to a footplate attached to a dual-mode muscle lever system (Model 129 300C-LR; Aurora Scientific, Aurora, Ontario, Canada). The knee was secured using a custom-made mounting system, and the ankle was positioned at a $90^o$ angle. First, under computer control of the servomotor, the ankle was passively rotated $20^o$ from neutral in both the plantar- and dorsi-flexion directions (total $40^o$ of motion). Second, to avoid recruitment of the anterior compartment muscles, the common peroneal nerve was severed. Platinum-Iridium percutaneous needle electrodes were placed across the sciatic nerve, which branches to the tibial nerve. Then optimal muscle stimulation was achieved by finding peak-isometric torque by increasing the current in increments of 0.2 mAmps. The force frequency relationship was assessed by measuring torque as a function of the following stimulation frequencies: 5, 10, 20, 40, 60, 80, 100, 150, and 200 Hz. Isometric torque is expressed as mN·m to determine absolute functional capacity or per kg body weight to assess functional quality.

## Biochemical analysis

At the terminal time points, the proximal and distal thirds of the gastrocnemius muscle of all animals were snap frozen using liquid nitrogen and subsequently stored at -80˚C until biochemical analysis. The distal third portion was weighed and homogenized in 10mM phosphate buffer (pH 7.4) at a ratio of 1:10 (mg/μl), using a glass pestle tissue grinder. Total protein content was analyzed in triplicate and averaged using the Protein A280 setting on a NanoDrop One spectrophotometer (Thermo Scientific). Mitochondrial content was analyzed by citrate synthase enzyme activity as previously described [16]. The proximal third portion of the gastrocnemius muscle remaining was homogenized in 33mM phosphate buffer (pH 7.4) at a muscle to buffer ratio of 1:40 using a glass tissue grinder. Homogenate was incubated with 5,5'-dithio-bis (2-nitrobenzoic acid (DTNB, 0.773 mM), acetyl CoA (0.116 mM), and oxaloacetate (0.441 mM) in 100 mM Tris buffer (pH 8.0). Activity of citrate synthase activity was monitored from the reduction of DTNB over time via measurement of absorbance at 412 nm. Succinate

dehydrogenase (SDH) activity was measured through the change of absorbance of cytochrome c. Muscle homogenate was sequentially incubated with 16.2 mM sodium succinate and 0.32 mM sodium cyanide. Then, was placed in assay buffer containing 0.327 mM aluminum chloride, 0.327 mM calcium chloride and 0.021 mM cytochrome c. SDH activity was monitored via the reduction of cytochrome c at 550 nm. Complex I activity was measured as previously described [20]. Briefly, muscle homogenate was incubated in 50 mM potassium phosphate buffer, supplemented with 3 mg/mL BSA, 240 μM KCN, 0.4 μM antimycin A, 50 uM decylubiquinone, and 80 μM 2,6-dichlorophenolindophenol (DCPIP). Nicotinamide adenine dinucleotide hydrogen (NADH) oxidation was measured through the reduction of DCPIP at 600 nm.

### Histological and morphologic analyses of muscle fibers

At harvest, the middle third of the gastrocnemius muscle of all animals was isolated for histological evaluation. Muscles were mounted on cork using tragacanth gum, frozen in 2-methylbutane cooled by liquid nitrogen, and subsequently stored at -80˚C. Ten μm sections of the mid-belly of the gastrocnemius muscle were collected using a Leica cryostat and microtome. Serial sections were stained first with Masson's Trichrome for qualitative analysis of skeletal muscle fibers and fibrotic deposition and quantification of centrally located nuclei. Subsequently, sections were stained to quantify capillarity, NADH-tetrazolium reductase and SDH to identify and quantify the proportion of oxidative muscle fibers, and myosin heavy chain (MyHC) fiber type composition.

Capillarity was assessed by staining for alkaline phosphatase activity [21]. Muscles were incubated in alkaline phosphate buffer for one hour at 37˚C. Sections were then washed, incubated with 0.25% metanil yellow for five minutes, and washed again. NADH-TR staining was performed by incubating tissues at 37˚C for 20 minutes in a solution containing 0.2M Tris, 1.5 mM NADH, and 1.5 mM NBT [22]. Sections were washed, dehydrated, and cleared in xylenes. SDH staining was performed by incubating tissues at 37˚C for one hour in a 0.2M sodium phosphate buffer solution [23]. Tissues were rinsed, dehydrated in graded acetone dilutions, and rinsed again.

Frozen muscles were also analyzed for MyHC isoform expression. Staining of the gastrocnemius muscle was completed with a combination of primary antibodies, including anti-MyHC$_{slow}$ (BA-D5, 5μg/ml), anti-MyHC$_{2A}$ (SC-71, 5μg/ml), and anti-MyHC$_{2B}$ (BF-F3, 5μg/ml) all acquired from the Developmental Hybridoma Studies Bank (Iowa City, IA). Additionally, sections were stained with wheat germ agglutinin (Invitrogen W7024, 1 μg/ml) to identify the sarcolemma. Appropriately paired secondary Alexa-Fluor (Invitrogen A21240) and DyLight (Jackson ImmunoResearch 115-475-207 or 115-545-020) conjugated antibodies at a dilution of 1:200 were used. In all cases, the expected staining patterns in normal skeletal muscle were observed and the specificity of anti-labeling was confirmed by the absence of staining outside expected structures and was consistent with manufacturer's technical information. The specificity of the MyHC isoform antibodies has been previously validated [24] and is representative of our previous work [25,26].

All brightfield images were acquired using the TissueScope LE slide scanner (Huron Digital Pathology, St. Jacobs, ON, Canada) using a 20X objective (0.75 NA, 0.5 μm/pixel resolution). Following imaging, using HuronViewer (Huron Digital Pathology), three standardized regions of interest (ROI) were selected for each muscle and exported for analysis. All ROIs were standardized to 500 x 500μm. One ROI was created to encompass the area immediately adjacent to the defect region of the VML group and two ROIs encompassed the remaining skeletal muscle tissue on medial and lateral side of the defect region. The corresponding three

ROI areas were also obtained in muscles of the control groups. All ROIs were systematically used across all muscles from the same animal. The ROIs were first evaluated independently within the respective region (lateral, mid-muscle, and medial). Subsequently, data from these ROIs were either summed or averaged across the gastrocnemius muscle and compared across groups.

Fluorescent imaging of all muscles stained for MyHC isoform expression was conducted using a Nikon C2 automated upright laser scanning confocal microscope equipped with a Plan Apo λ 20x objective and dual GaSP detectors (Nikon Instruments Inc., Melville, NY). Image sampling was determined using Nyquist calculations with a pixel size set to 0.31μm and pixel dimension of 2048 x 2048. Images were collected from the same three ROIs previously described. All ROIs were subsequently saved and exported for analysis.

Analyses of muscle sections were conducted using Fiji [27]. The multipoint tool was used to manually count muscle fibers, centrally located nuclei, and capillaries, as well as the number of darkly and lightly stained muscle fibers for all NADH and SDH stained muscle. Darkly stained fibers were classified as positive for NADH and SDH, indicating high metabolic activity, while lightly stained fibers were classified as negative for NADH and SDH, suggesting low metabolic activity. Fibers of the gastrocnemius muscle were classified signally as type I, IIa, and IIb based on the expression of $MyHC_{slow}$, $MyHC_{2A}$, and $MyHC_{2B}$, respectively. Fibers were classified as type IIx based on the absence of staining. Additionally, muscle fiber cross-sectional area (CSA) was measured for each section using the freehand tool and quantified as fiber-type specific CSA. Muscle CSA was only quantified for fibers between 50–7,500μm$^2$. For display purposes only, images were down-converted, without introducing any changes in brightness or contrast and produced in Adobe Photoshop (Adobe Systems Inc.). During all imaging, laser intensity was kept consistent across imaging of the same probes. Investigators were blinded during all imaging and post-imaging analyses. If samples did not have all three regions of interest available for analysis, they were excluded from combined analyses but still included in regional evaluations.

## Statistical analyses

Data analysis was conducted using JMP (version 14.3.0, SAS Institute, Inc.) and Prism (version 9, GraphPad Inc). For metabolic and physical activity data, one-way ANOVAs were used to evaluate differences across time frames of 1-, 6-, 12-, and 24-hr periods. Longitudinal differences across time, pre-VML and 2- and 6-weeks post-VML were calculated using means and AUC, differences were evaluated using one-way, repeated measures, ANOVA. One-way ANOVAs were used to evaluate differences across groups for body weight, muscle weight, muscle function, histologic parameters, and protein and enzyme content. Two-way ANOVAs were used to evaluate overall muscle fiber-type specific differences across group. Distributions of capillaries per muscle fiber and fiber CSA were evaluated by chi-squared, and alpha was corrected for multiple comparisons at an alpha of 0.0167. When appropriate Tukey's HSD post hoc was evaluated, significance was set at an alpha of 0.05. Data is reported as mean ± SD unless noted otherwise.

## Results

### Animals

During the pre-VML metabolic and physical activity evaluation, mice weighed 27.7±2.0g. Following all surgical procedures mice recovered promptly and no adverse events were identified. As expected, mice gained weight over the 4- and 8-week period, with an average terminal body

**Table 1. Contractile properties following VML injury.**

|  | Control | VML 4-week | VML 8-week | p-value |
|---|---|---|---|---|
|  | (n = 16) | (n = 5) | (n = 7) |  |
| Body weight (g) | 30.5 ± 0.6 | 29.3 ± 1.1 | 31.1 ± 0.9 | 0.427 |
| Gastrocnemius weight ratio | 1.08 ± 0.08 | 0.84 ± 0.07 * | 0.92 ± 0.15 * | <0.001 |
| Passive torque (mN·m) | 2.4 ± 0.5 | 4.2 ± 1.4 * | 5.1 ± 1.4 * | <0.0001 |
| Peak twitch 5Hz (mN·m) | 4.1 ± 0.9 | 2.9 ± 0.7 * | 1.9 ± 1.0 * | <0.0001 |
| Peak torque (mN·m) | 19.2 ± 2.5 | 12.3 ± 1.2 * | 9.9 ± 2.4 * | <0.0001 |
| Twitch:tetani ratio | 0.21 ± 0.01 | 0.23 ± 0.02 | 0.19 ± 0.02 | 0.313 |

Data analyzed by one-way ANOVA, presented as mean ± SD

*different than control.

mass of ~30 and 31g, respectively (Table 1). Body weight was not significantly different between groups at any time during the study duration (p = 0.123).

## Baseline physical and metabolic activity

Prior to allocation into experimental groups, all mice were analyzed for 24-hour physical and metabolic activity. At this baseline time point, mice ambulated 1.3±0.3km over 24 hours, with ~1km ambulated during the active period (i.e., dark hours of 18:00–06:00) and the remaining over the inactive period (i.e., light hours of 06:00–18:00). Over the course of 24 hours, the average metabolic rate was ~19 kcal/hr. As expected, metabolic rate was greatest during the active period, reaching the highest value of 23.2±2.3 kcal/hr at 20:00, and lesser during the inactive period, reaching the lowest value of 15.6±1.8 kcal/hr at 10:00 (p<0.001; Fig 1A). Diurnal metabolic rate flux, as quantified by the delta metabolic rate between phases, was ~4.5kcal/hr. Metabolic rates fluctuated between 19–24 kcals/hr during the active period and were greatest early in the active period, specifically from 18:00–22:00 (p<0.001; Fig 1E). During the inactive period, metabolic values fluctuated between 15–19 kcal/hr with the greatest values occurring during the transition from active to inactive, around 05:00–08:00 hours. Hourly metabolic rate was typically below 17 kcals/hr during the rest of the inactive time period (p<0.001).

The RER over the course of 24 hours was ~0.90, with a RER of ~0.93 during the active period but only ~0.87 during the inactive period (p<0.001; Fig 1B). At baseline the 24-hour RER corresponded to a carbohydrate oxidation of 5.87±0.91 g/min and lipid oxidation of 0.95±0.25 g/min. The greatest RER during the active period was 0.95±0.04 at 21:00, and the lowest RER value during the inactive period was 0.84±0.04 at 10:00. This range of RER values served as a baseline level of metabolic flexibility for comparison and supported by a greater carbohydrate oxidation during the active period (p<0.001; Fig 1C), and greater lipid oxidation during the inactive period (p<0.001; Fig 1D). Hourly RER fluctuated between 0.90–0.96 during the active period, with a gradual decline during the transition to the inactive period from 05:00–11:00. Inactive RER values fluctuated between 0.84–0.93, with the lowest values occurring during the middle of the active period from 11:00–13:00, then gradually increasing through 17:00 (p<0.001; Fig 1G).

## VML-induced changes in physical and metabolic activity

Following VML injury both physical and metabolic activity were longitudinally evaluated in a subset of mice at 2- and 6-weeks post-VML surgery. Opposing the hypothesis, there was no difference in physical activity, specifically ambulatory distances, longitudinally after VML injury. The total distances ambulated for pre-, 2- and 6-weeks post-VML were 1.3±0.3, 1.4

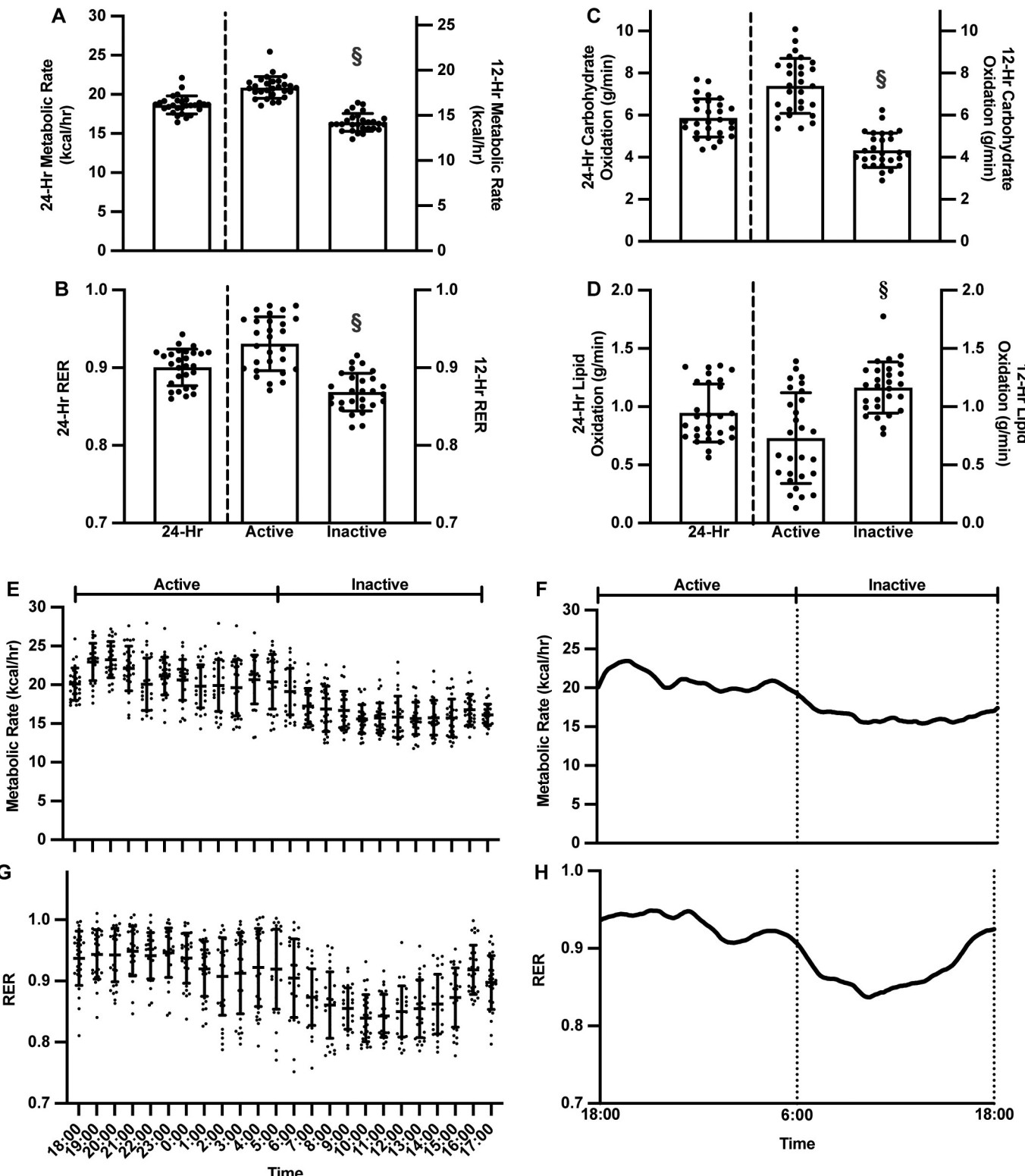

**Fig 1. 24-hour whole-body metabolism and physical activity prior to group allocation.** Mice were housed in a 12-hour light:dark environment matching the inactive and active time periods, respectively. Metabolic rate and respiratory exchange ratio (RER) were analyzed in 1-, 12-, and 24-hour bins. **A)** Metabolic rate was highest during the active period, which was significantly greater than during the inactive period (p<0.001). **B)** RER was highest during the active period. **C)** Reflective of RER, carbohydrate oxidation was higher during the active period than during the inactive period (p<0.001) while **D)** lipid oxidation was higher during the inactive period than the active period (p<0.001). **E)** In 1-hr bins the greatest metabolic rate recorded during the active period. **F)** The total area under

the curve (AUC) for metabolic rate was 5,936±176. **G)** In 1-hr bins the greatest RER occurred during the active period (p<0.001) and **H)** RER followed the same trend as metabolic rate, with an AUC of 63.8±3.2. Data presented as mean±SD; analyzed by one-way ANOVA; significantly different from §active period.

±0.2, and 1.4±0.4km, respectively (p = 0.809). As expected, most of the ambulatory activity occurred during the active time period.

Longitudinal metabolic activity measurements supported the hypothesized decrease in metabolic rate by 6-weeks post-VML. Specifically, the 24-hour metabolic rate decreased ~10% compared to pre- and 2-weeks post-VML (p = 0.012; Fig 2A). Similarly, metabolic rate was decreased ~9% by 6-weeks post-VML during the active period compared to 2-weeks post-VML (p = 0.032) and was reduced ~12% by 6-weeks compared to pre- and 2-weeks post-VML, during the inactive period (p = 0.092; Fig 2A). There was no change in delta metabolic rate, i.e., the difference in metabolic rate between the active and inactive periods, at any time point evaluated, 4.9±1.0, 4.8±0.6, and 4.8±1.3 at pre-, 2-weeks and 6-weeks post-VML, respectively (p = 0.988). To more specifically evaluate possible differences in metabolic flexibility, metabolic rate was also compared across time points with the calculated AUC. The 24-hour metabolic rate AUC was significantly decreased by 6-weeks post-VML (5,400±30) compared to pre- and 2-weeks post-VML (5,948±33 and 6,026±30, p<0.001; Fig 2B). Subsequently, the AUC was evaluated in 6-hour bins. Within each active 6-hour bin, the metabolic rate 6-weeks post-VML was significantly reduced compared to the pre-VML and 2-weeks post-VML time points (p≤0.001; Fig 2B). Only during the second phase of the inactive period (00:00–06:00) was the metabolic rate at 2-weeks post-VML different from pre-VML (p = 0.007; Fig 2B).

In addition to metabolic rate, RER was evaluated longitudinally. At 6-weeks post-VML 24-hour RER was significantly reduced by ~4% compared to the pre-VML (p = 0.014), due to the reduction in RER during the active time (p = 0.003; Fig 2C). The decrease in RER resulted from a significant decrease in carbohydrate oxidation and concomitant increase in lipid oxidation over the course of 24 hours and during the active period (p≤0.002; Fig 2E and 2F). There was no difference in the delta RER over time, with delta values of 0.09±0.02, 0.08±0.3, and 0.04±0.04 at pre-, 2-weeks and 6-weeks post-VML, respectively (p = 0.530). The 24-hour metabolic flexibility in RER was significantly lower at 2- and 6-weeks post-VML compared to pre-VML (AUC 63.5±4.0, 59.8±4.4 vs. 70.7±4.2, p<0.001; Fig 2D). The RER during the active period was greatest pre-VML, and lowest at 6-weeks post-VML, particularly during the first 6 hours of the active period (p<0.001; Fig 2D). A similar pattern was observed during the inactive period, in which the RER pre-VML was the greatest, while at 6-weeks post-VML RER was the lowest, especially during the second phase of the inactive period (p<0.001; Fig 2D). Collectively, indicating metabolic inflexibility up to 6-weeks following VML injury.

### *In vivo* muscle function

The VML injury to the posterior compartment resulted in an ~22% reduction of gastrocnemius mass at 4-weeks, and ~15% reduction at 8-weeks (p<0.001; Table 1). The passive torque at 20° of dorsiflexion was determined as a surrogate of passive muscle stiffness. The VML injury resulted in a significant increase in passive torque about the ankle joint by ~2 fold at 4-weeks and 8-weeks (p<0.001; Table 1). The VML injury did not impact the twitch to tetani ratio of the ankle plantar flexor muscles (Table 1). There was no difference in the force-frequency (1–300 Hz) relationship of the ankle plantar flexor muscles, maximal torque was reached at a frequency of 200 Hz, and the ~50% torque deficit in the VML group was observed at various frequencies from 40–80 Hz. Compared to controls, maximal isometric torque of the ankle plantar flexor muscles decreased by about 50% at both 4-weeks and 8-weeks post-VML

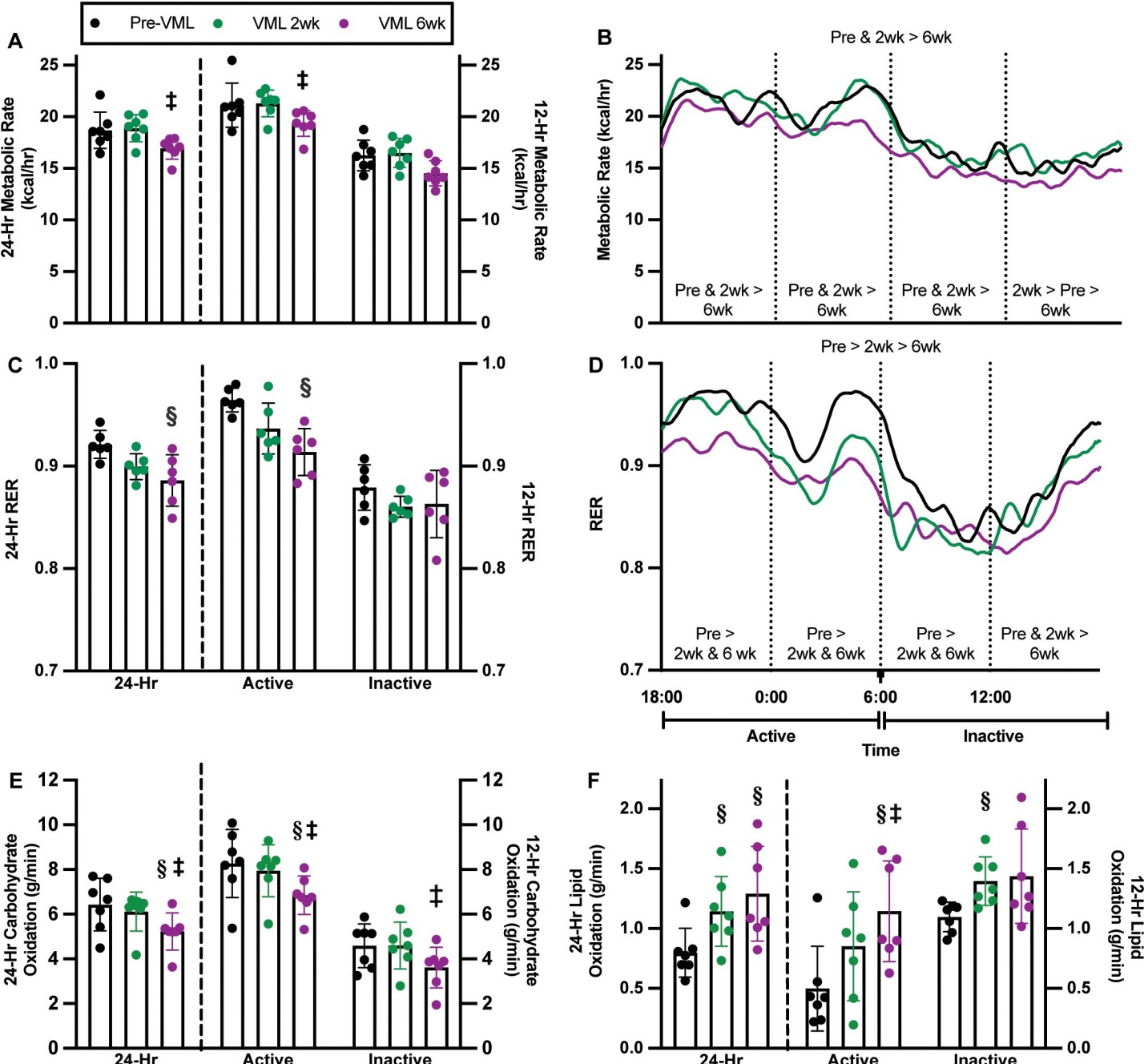

**Fig 2. Longitudinal evaluation of metabolic rate and RER, pre-VML surgery and at 2- and 6-weeks post-VML, in a subset of mice. A)** By 6-weeks post-VML there was decreased metabolic rate during the 24-hour and 12-hour active periods, compared to 2-weeks post-VML (p≤0.050). **B)** Metabolic rate was evaluated by comparing AUC values across time. When broken into 6-hour bins, significant differences were noted across time points within each bin, with mice at 6-weeks post-VML demonstrating the lowest metabolic rate in each 6-hour bin (p≤0.007). **C)** The RER at 6-weeks post-VML decreased over 24 hours compared to pre-VML (p = 0.013), likely due in part to a significant reduction in RER which occurred during the active period by 6-weeks post-VML (p = 0.002). **D)** At 2-weeks post-VML, mice exhibited a lower RER as determined by AUC compared to pre-VML, and a further significant reduction in RER at 6-weeks post-VML, indicating progressively impaired metabolic flexibility 2- and 6-weeks post-VML compared to baseline (p<0.001). In 6-hour bins, significant differences were observed across evaluation time points for each bin (p≤0.001). **E)** Twenty-four-hour carbohydrate oxidation 6-weeks post-VML was significantly reduced, primarily due to decreases during the active period (p = 0.002). **F)** Lipid oxidation over the course of 24 hours was significantly increased 6-weeeks post-VML (p<0.001) due to significantly increased oxidation during the active period. Data presented as mean±SD; analyzed by one- or two-way ANOVA; significantly different from §pre-VML; ‡VML 2wk.

**Table 2. Biochemical composition of gastrocnemius muscle following VML injury.**

|  | Control | VML 4-week | VML 8-week | p-value |
|---|---|---|---|---|
|  | (n = 16) | (n = 5) | (n = 7) |  |
| Total protein content (mg/mg muscle) | 20.4 ± 3.4 | 19.3 ± 3.6 | 18.7 ± 2.4 | 0.674 |
| Citrate synthase (μmol/min/g) | 498.8 ± 75.4 | 458.0 ± 77.8 | 480.3 ± 116.3 | 0.444 |
| Complex I (μmol/min/g) | 16.4 ± 5.4 | 10.9 ± 4.7 | 17.9 ± 7.3 | 0.116 |
| Succinate dehydrogenase activity (μmol cyto c/min/g) | 4.93 ± 0.68 | 4.12 ± 0.95 | 4.86 ± 0.80 | 0.128 |

Data analyzed by one-way ANOVA, presented as mean ± SD.

(p<0.001; Table 1). Both absolute and torque normalized to body mass were significantly impaired by VML injury.

## Biochemical analysis of gastrocnemius muscles

The remaining gastrocnemius muscle had minimal VML-induced changes to total protein content (p = 0.674). Similarly, citrate synthase activity, a surrogate of mitochondrial abundance, was also unchanged (p = 0.444). There were also no differences in Complex I or succinate dehydrogenase activity (p≥0.128). Collectively, biochemical aspects of the gastrocnemius muscle remaining after injury appear largely unchanged post-VML at the time points evaluated (Table 2).

## Histological analysis of gastrocnemius muscles

Whole muscle and regional histologic characteristics of gastrocnemius muscles were evaluated terminally across groups. In comparison to control muscles, qualitative examination indicated fibrotic development in muscles post-VML, with some muscle fibers that appeared smaller in size and irregular in shape. There was also an abundance of centrally located nuclei, particularly in the medial portion of the muscle near the initially created VML injury (Fig 3). The percentage of fibers with centrally located nuclei was greater in the medial and mid-gastrocnemius regions at 4-weeks post-VML and in all regions at 8-weeks post-VML (p≤0.013; Table 3).

Specific changes in the VML-injured gastrocnemius muscle fibers were quantified across regions of the whole muscle (Table 3). The lateral and medial regions did not experience a change in the number of capillaries per fiber (p≥0.345), while the mid-region bordering the VML defect demonstrated a reduction in capillaries at 4-weeks, and an increase at 8-weeks post-VML (p = 0.041). Across the entire gastrocnemius muscle, there was a difference in the distribution of capillaries per muscle fiber muscle at both 4- and 8-weeks post-VML compared to control (p<0.001). Between the 4- and 8-weeks post-VML groups there was also a leftward shift in capillary distribution (p = 0.001), in parallel with an increased proportion of capillaries per fiber at the tail ends of the distribution curve (Fig 4A). Across the whole gastrocnemius muscle there were no differences between groups in the number of capillaries per fiber (p = 0.154; Fig 4B). At 8-weeks post-VML, there was a greater percentage of central nuclei across the whole muscle compared to controls (p = 0.007), but a similar percentage to muscles at 4-weeks post-VML (p = 0.395; Fig 4C). There were no differences in central nuclei percentages between muscles at 4-weeks post-VML and controls (p = 0.338; Fig 4C).

In examining the oxidative capacity of the muscle, expression of NADH-positive fibers was only different 4-weeks post-VML in the medial region compared to control (p = 0.036), while no differences in SDH-positive fibers were observed in any region at any time point (p≥0.163;

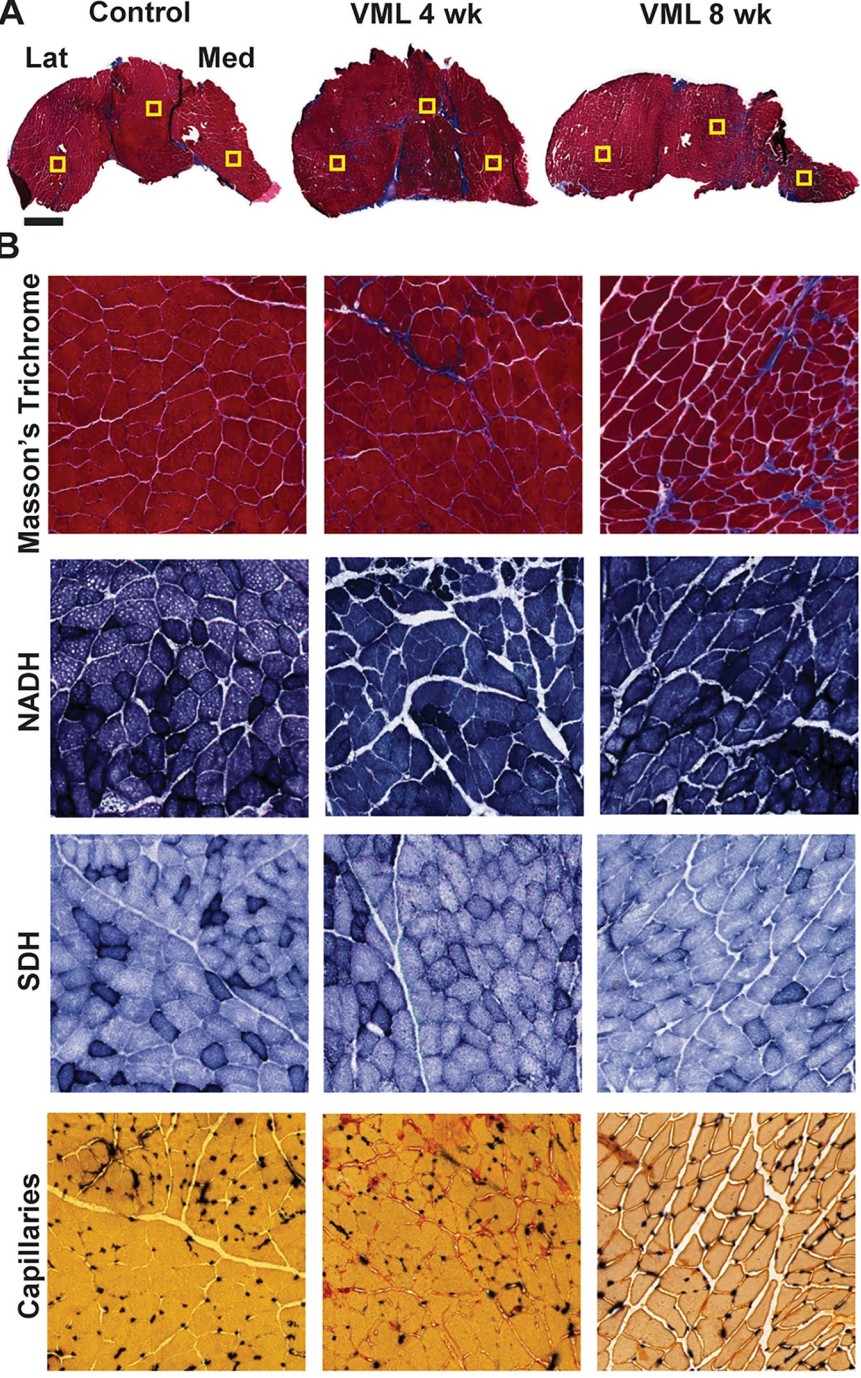

**Fig 3. Histologic evaluation of the gastrocnemius muscle following VML. A)** Representative whole gastrocnemius muscles for each group. Three standardized regions of interest were evaluated for each muscle (boxes), representing the lateral, middle/defect, and medial aspects of the gastrocnemius muscle. Scale bar is 1mm. **B)** Serial sections of the gastrocnemius muscle were stained for various physiologic and oxidative characteristics, and representative images from the middle/defect region are shown. Masson's Trichrome (muscle fibers red; fibrosis blue; nuclei black) was used to evaluate muscle quality and centrally located nuclei. NADH and SDH were used to evaluate oxidative enzymes, with darker stained fibers indicating fibers positive for NADH or SDH, respectively. Alkaline phosphatase was used to evaluate fiber capillarity (muscle fibers yellow; capillaries black). Scale bar is 100μm.

**Table 3. Histologic composition of the gastrocnemius muscle following VML.**

|  | Control | VML 4-week | VML 8-week | p-value |
|---|---|---|---|---|
| *Lateral Gastrocnemius* | | | | |
| Fibers with centrally located nuclei (%) | 7.2 ± 8.2 | 20.2 ± 9.6 | 33.1 ± 31.6 * | 0.013 |
| Mean capillaries per fiber | 5.6 ± 1.1 | 4.4 ± 1.0 | 4.8 ± 1.1 | 0.698 |
| NADH positive fibers (%) | 75.1 ± 11.8 | 87.2 ± 2.7 | 85.0 ± 11.2 | 0.088 |
| SDH positive fibers (%) | 53.7 ± 20.2 | 72.5 ± 19.5 | 67.7 ± 16.8 | 0.163 |
| *Mid-Gastrocnemius; VML border* | | | | |
| Fibers with centrally located nuclei (%) | 1.6 ± 2.3 | 28.1 ± 14.1 * | 18.7 ± 9.2 * | <0.001 |
| Mean capillaries per fiber | 3.6 ± 1.1 | 2.7 ± 1.0 * | 3.9 ± 1.1 * | 0.041 |
| NADH positive fibers (%) | 63.0 ± 18.0 | 78.7 ± 13.3 | 72.8 ± 15.9 | 0.202 |
| SDH positive fibers (%) | 30.6 ± 21.5 | 34.0 ± 23.6 | 25.8 ± 13.4 | 0.788 |
| *Medial Gastrocnemius* | | | | |
| Fibers with centrally located nuclei (%) | 8.2 ± 12.8 | 21.8 ± 16.3 * | 29.7 ± 13.9 * | 0.005 |
| Mean capillaries per fiber | 4.6 ± 1.1 | 5.3 ± 0.6 | 5.2 ± 1.2 | 0.345 |
| NADH positive fibers (%) | 71.5 ± 15.5 | 91.3 ± 1.5 * | 59.6 ± 21.2 | 0.036 |
| SDH positive fibers (%) | 58.0 ± 13.8 | 66.4 ± 19.2 | 58.0 ± 16.0 | 0.663 |

Data analyzed by one-way ANOVA, presented as mean ± SD

*different from control.

Table 3). Across the whole muscle, there was an increase in NADH-positive muscle fibers at 4-weeks post-VML compared to control but not compared to 8-weeks post-VML (p≥0.022). No differences in the percentage of NADH-positive fibers were observed between controls and 8-weeks post-VML (Fig 4D). There were no differences in percentage of SDH-positive muscle fibers across control, 4- or 8-weeks post-VML (p≥0.278; Fig 4E).

Gastrocnemius muscles were evaluated as a whole and regionally for fiber type specific differences in proportions and fiber CSA following VML injury. Regionally, no differences in CSA were observed between post-VML and control groups (Table 4). Across the whole muscle, the distribution of muscle fiber CSA shifted leftward at 4- and 8-weeks post VML compared to control (p<0.001), and VML groups were not significantly different from each other (p = 0.017 notably non-significant due to multiple comparisons correction; Fig 5A). A main effect of experimental group was for fiber type specific CSA, where smaller CSA were observed 4-weeks and 8-weeks post-VML compared to controls (p<0.001). A main effect of fiber type was also observed across the whole muscle, whereby type IIb fibers were larger than type I, IIa, and IIx fibers, and type IIx fibers were larger than type I and IIa fibers (all p<0.001; Fig 5B). When proportions of fiber types were compared across the muscle, there was a significant interaction between group and fiber type (p<0.001), where type I, IIx, and IIa were the least represented, and type IIb was the most represented. Compared to control, VML groups tended to have greater expression of type I and IIa fibers, and lower type IIb, particularly at 4 weeks (Fig 5C). Regionally, compared to control, the greatest fiber type specific differences occurred in the medial portion of the gastrocnemius muscle at 4-weeks post-VML, with VML injured muscles demonstrating higher proportions of type I and type IIa fibers and smaller proportions of type IIb fibers (Table 5 and Fig 5D).

## Discussion

The pathophysiologic comorbidities of VML injury are vast and range from functional deficits, impaired inflammatory signaling, denervation, and chronic fibrotic deposition [28,29].

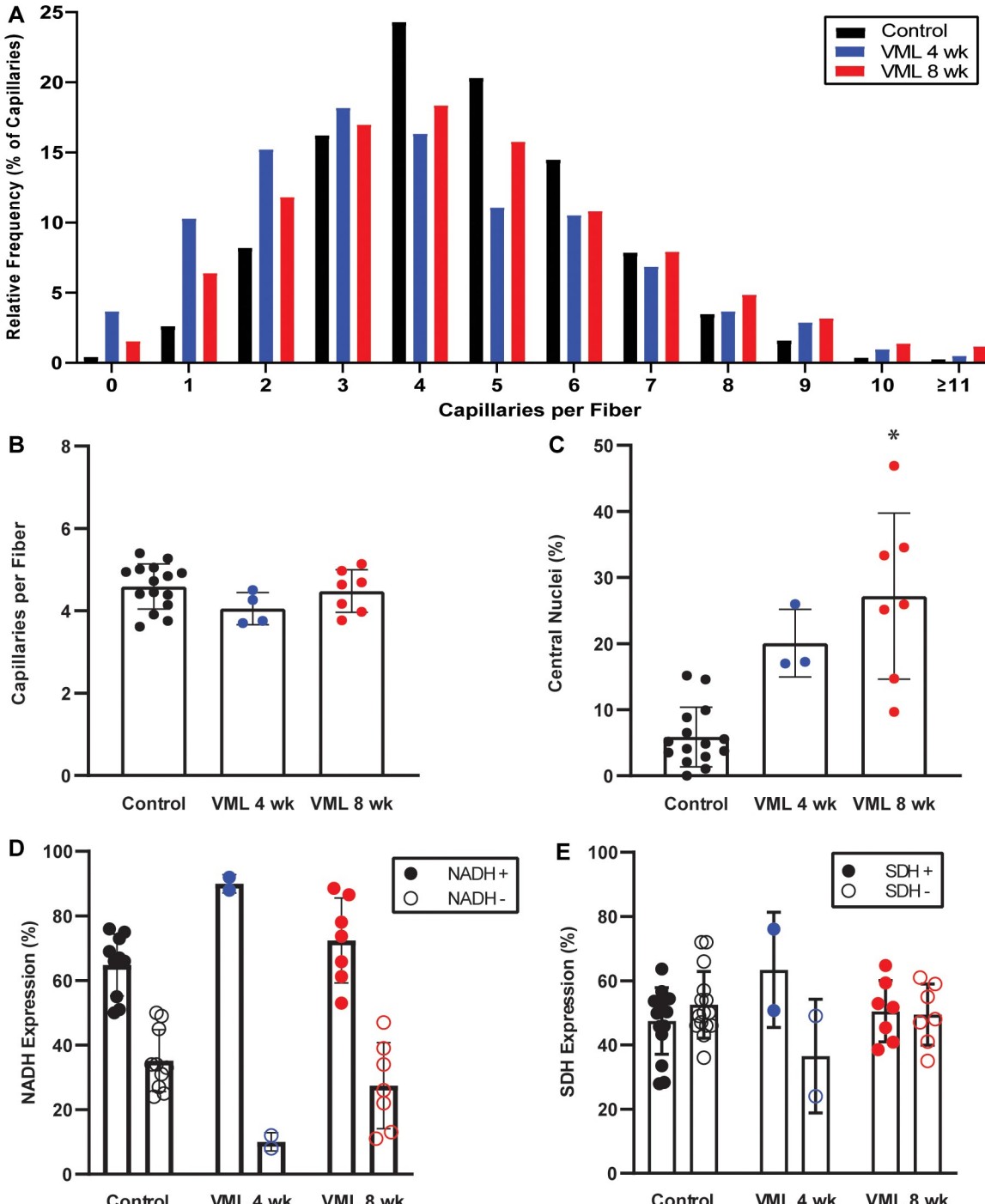

**Fig 4. Muscle-fiber specific histology in the gastrocnemius muscle following VML. A)** The distribution of capillaries was significantly different at 4- and 8-weeks post-VML compared to controls (p<0.001) and between groups at 4- and 8-weeks post-VML (p<0.001). **B)** There were no differences in the average number of capillaries per gastrocnemius muscle fiber (p = 0.154). **C)** Compared to control, muscles at 8-weeks post-VML had a greater percentage of centrally located nuclei (p = 0.007). **D)** There was a significant increase in NADH-positive fibers 4-weeks post-VML compared to control (p = 0.016) but not compared to 8-weeks post-VML (p = 0.059). **E)** There were no significant differences between control, 4- or 8-weeks post-VML (p = 0.625). Data presented as mean±SD; analyzed by chi-squared and one-way ANOVA; significantly different from *control.

**Table 4. Fiber type specific size of gastrocnemius muscle following VML.**

| | Control | VML 4-week | VML 8-week | p-value |
|---|---|---|---|---|
| *Lateral Gastrocnemius* | | | | |
| Type I (mean CSA μm$^2$) | 1333.5 ± 249.4 | 1232.6 ± 431.9 | 1454.0 ± 334.6 | 0.917 |
| Type IIa (mean CSA μm$^2$) | 1416.9 ± 142.2 | 1278.0 ± 229.2 | 1530.7 ± 229.22 | 0.740 |
| Type IIb (mean CSA μm$^2$) | 3026.7 ± 239.8 | 2218.1 ± 428.95 | 2467.6 ± 391.6 | 0.205 |
| Type IIx (mean CSA μm$^2$) | 2211.7 ± 231.1 | 1906.3 ± 386.7 | 1771.9 ± 353.0 | 0.545 |
| *Mid-Gastrocnemius; VML border* | | | | |
| Type I (mean CSA μm$^2$) | 1099.1 ± 163.4 | 680.6 ± 346.6 | 1009.5 ± 219.2 | 0.564 |
| Type IIa (mean CSA μm$^2$) | 1289.0 ± 148.7 | 1138.4 ± 284.7 | 1563.3 ± 186.4 | 0.383 |
| Type IIb (mean CSA μm$^2$) | 2977.7 ± 238.3 | 1792.7 ± 426.3 | 2595.0 ± 360.3 | 0.070 |
| Type IIx (mean CSA μm$^2$) | 1777.9 ± 179.7 | 1268.7 ± 311.3 | 1976.9 ± 235.3 | 0.213 |
| *Medial Gastrocnemius* | | | | |
| Type I (mean CSA μm$^2$) | 834.7 ± 175.0 | 846.1 ± 234.7 | 1150 ± 234.7 | 0.535 |
| Type IIa (mean CSA μm$^2$) | 1241.6 ± 104.1 | 1104.1 ± 180.3 | 1165.6 ± 164.5 | 0.787 |
| Type IIb (mean CSA μm$^2$) | 2746.3 ± 219.5 | 2393.5 ± 392.6 | 2128.7 ± 331.8 | 0.296 |
| Type IIx (mean CSA μm$^2$) | 1597.4 ± 188.7 | 1641.4 ± 337.6 | 1791.3 ± 308.2 | 0.867 |

Data analyzed by one-way ANOVA, presented as mean ± SD.

However, to date both muscle-specific and whole-body metabolic consequences of VML injury are poorly understood, both in animal models and the patient population. The most salient finding herein is that even with a relatively small volume of the total body skeletal muscle lost to VML, there are substantial local and whole-body impairments to metabolism. Whole-body metabolism alterations seem to be largely due to changes in carbohydrate and lipid utilization that occur during active hours, rather than reductions in overall physical activity. Consequentially, chronic impairments in both muscle-specific and whole-body metabolism could confound other comorbidities in the sequela of VML.

Following VML, there was no difference in body mass and physical activity was not reduced. Supportive of previous work in which VML-injured rodents demonstrate similar voluntary running distances to uninjured rodents [16,30]. Despite the lack of change in physical activity, whole-body metabolic activity was significantly impaired, quantified by progressive decreases in metabolic rate and RER. Reduced RER resulted from lower carbohydrate and greater lipid utilization primarily within the active period. It is known that diurnal metabolic outcomes, particularly during the active period, are influenced by physical activity and eating habits [31]. Although both physical activity and body mass did not change, the thermal effect of eating and postprandial RER cannot be ruled out as partially influencing active period metabolic observations herein.

Other factors besides physical activity and eating may play a role in metabolic changes, such as hormonal and mitochondrial dysregulation. Clinically, mitochondrial dysregulation is evident in patients suffering from burns, neurodegenerative disorders, insulin resistance, and aging, among other conditions [32–36]. These pathologies can lead to metabolic inflexibility and increased risk for additional metabolic and cardiovascular comorbidities. In comparative rodent models, the development of insulin resistance (hormone dysregulation) coincides with a decrease in RER and excessive β-oxidation, suggesting fatty acids infiltrate the muscle mitochondria [37]. In studies of acute burn injury, mitochondrial dysregulation presents as a hypermetabolic state concurrent with significant declines in skeletal muscle ATP production rates [35]. Mitochondrial inflexibility results, in part, from mitochondrial uncoupling, a

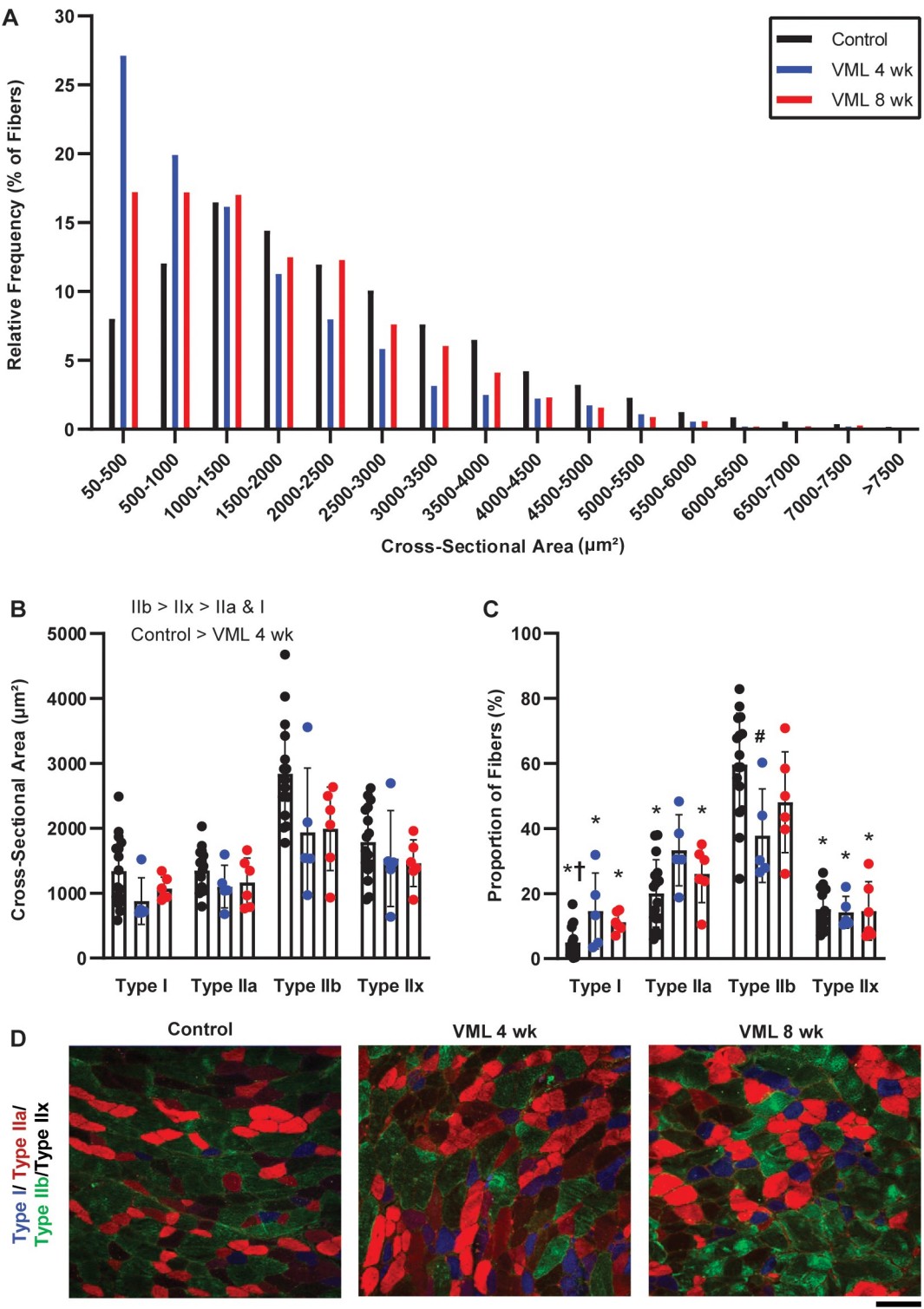

**Fig 5. Fiber type specific differences in the gastrocnemius muscles following VML. A)** Muscle fiber cross-sectional area (CSA) in controls was normally distribute, in comparison the distribution of muscle fiber CSA shifted left at 4- and 8-weeks post-VML (p<0.001), but VML groups were not different than each other (p<0.017). **B)** Overall muscle fiber CSA was smaller at 4-weeks post-VML compared to control (main effect experimental group; p = 0.011). As expected, type IIb fibers had a greater CSA than type I, IIa, and IIx; additionally, type IIx fibers were larger than type I and IIa fibers (main effect of fiber type; p<0.001). **C)** There was a significant group by fiber type interaction (p = 0.001) in the proportion of fiber types across the gastrocnemius muscle; control and post-VML muscle demonstrated a higher proportion of type IIb fibers

compared to other muscle fiber types. Control muscles demonstrated a greater proportion of type IIb fibers compared to muscles 4-weeks post-VML. **D)** Myosin heavy chain (MyHC) isoform expression was evaluated to determine proportions of type I, IIa, IIb, and IIx fibers, staining blue, red, green, or black, respectively. Scale bar is 100μm. Data presented as mean±SD; analyzed by chi-squared and two-way ANOVA, significant main effects are indicated within figure. Within experimental group, significantly different from *type IIb; † type IIa. Within fiber type, significantly different from #control.

dissociation between mitochondrial membrane potential generation and its use for mitochondria-dependent ATP production [38]. Mild uncoupling may hinder the ability of the mitochondria to meet energy demands of the muscle following VML injury and severe uncoupling may lead to autophagy activation, reactive oxygen species production, maladaptive cell signaling, and cell death [38].

Examination of muscle-specific metabolism and oxidative capacity can provide insight into the metabolic inflexibility observed herein. For example, the greater NADH histological staining could indicate: i) hypoxia-induced decline in respiration and less complex I oxidation of NADH, ii) greater cellular dehydrogenase activity leading to an accumulation of NADH (i.e., increased β-oxidation), and/or iii) hyperpolarization of the mitochondrial membrane and a bioenergetic back-pressure on complex I activity. Oxygen is the second most electronegative element and provides the electrical "pull" for electrons through the electron transport chain. Mitochondrial dysregulation following VML has been quantified as a reduction in muscle fiber oxidative capacity, and by measures of vascularity [15–17]. Herein, decreased capillary distribution across the muscle following VML suggests the muscle may have a decreased supply of blood and oxygen, perhaps resulting from a pro-fibrotic microenvironment and leading to hypoxia [39–42]. Hypoxia can impair oxidation of NADH to $NAD^+$ within the electron transport chain, preventing adequate electron transport across the inner mitochondrial membrane to drive ATP production. Hypoxia can also inhibit pyruvate dehydrogenase activity through HIF-1α, leading to increased reliance on lipid oxidation for ATP production [43].

If a greater reliance on lipid oxidation does exist chronically due to VML (supported by RER data herein), then this can influence metabolic fuel selection and other critical aspects of

**Table 5. Fiber type specific composition of gastrocnemius muscle following VML.**

| | Control | VML 4-week | VML 8-week | p-value |
|---|---|---|---|---|
| *Lateral Gastrocnemius* | | | | |
| Percent of Type I Fibers | 3.9 ± 1.9 | 9.1 ± 3.5 | 15.8 ± 3.2 * | 0.012 |
| Percent of Type IIa Fibers | 20.4 ± 4.0 | 37.7 ± 7.1 | 32.6 ± 6.5 | 0.078 |
| Percent of Type IIb Fibers | 60.4 ± 6.5 | 35.8 ± 11.7 | 36.9 ± 10.6 | 0.083 |
| Percent of Type IIx Fibers | 15.2 ± 2.5 | 17.4 ± 4.4 | 14.7 ± 4.0 | 0.888 |
| *Mid-Gastrocnemius; VML border* | | | | |
| Percent of Type I Fibers | 7.2 ± 3.5 | 15.7 ± 6.3 | 6.1 ± 5.3 | 0.451 |
| Percent of Type IIa Fibers | 19.2 ± 4.3 | 22.8 ± 7.7 | 26.5 ± 6.5 | 0.640 |
| Percent of Type IIb Fibers | 59.7 ± 7.6 | 54.7 ± 13.7 | 50.0 ± 11.5 | 0.777 |
| Percent of Type IIx Fibers | 13.9 ± 2.6 | 6.8 ± 4.7 | 17.4 ± 4.0 | 0.234 |
| *Medial Gastrocnemius* | | | | |
| Percent of Type I Fibers | 3.9 ± 2.3 | 19.1 ± 4.1 * | 13.4 ± 3.4 | 0.006 |
| Percent of Type IIa Fibers | 20.6 ± 3.4 | 39.5 ± 6.1 * | 22.9 ± 5.2 | 0.039 |
| Percent of Type IIb Fibers | 59.0 ± 22.9 | 22.8 ± 10.1 * | 50.7 ± 8.5 | 0.016 |
| Percent of Type IIx Fibers | 16.6 ± 2.3 | 18.6 ± 4.2 | 13.0 ± 3.6 | 0.574 |

Data analyzed by one-way ANOVA, presented as mean ± SD

*different than control.

metabolism such as reactive oxygen species production, redox balance, altered mitochondrial membrane permeability and membrane potential [43]. For example, an increase in lipid oxidation produces a greater amount of β-oxidation supported Acetyl-CoA and citrate. Both Acetyl-CoA and citrate can inhibit key enzymes related to glucose metabolism like pyruvate dehydrogenase and phosphofructokinase. Left unchecked, this can produce a metabolic gridlock [44] in which even under cellular conditions favorable for glucose metabolism, the cell remains resilient to glucose oxidation because of enzymatic inhibition. Moreover, greater production of NADH and $FADH_2$ from β-oxidation increases delivery of these cofactors to the electron transport chain above that produced from the tricarboxylic acid cycle. However, if increased ATP production is not required due to sustained levels of oxidative phosphorylation, cofactors accumulate, and a greater NADH to NAD+ ratio is favorable to reactive oxygen [45]. Observations of mitochondrial dysregulation and uncoupling have also been noted following burn injury [35], denervation [46,47], immobilization [48], and in obesity [37]. Similarly, reports of a greater reliance on lipid oxidation has been noted in models of pathological conditions, including diabetic cardiomyopathy, heart failure, and following ischemia-reperfusion [49], amyotrophic lateral sclerosis [50], and other neurologic disorders [51]. The ratio of NADH to $FADH_2$ oxidation also reflects mitochondrial function and depends on the proton motive force required to drive ATP production, whereby hyperpolarization and reduced proton motive force by uncoupling increases $FADH_2$ oxidation, reflected in increased lipid oxidation [52]. Metabolic gridlock, redox imbalance, increased cofactor production, and altered ratios of cofactor oxidation could partially explain the mitochondrial dysfunction and metabolic inflexibility following VML; however, more work is necessary to understand these complex relationships in models of VML injury.

As expected, a chronic loss of muscle function was observed following VML. While primary mechanisms of VML-induced force loss are the loss of contractile tissue and endogenous regenerative capacity, other mechanisms are expected to contribute to this chronic dysfunction but are currently unclear. Alteration in fiber type proportions is a more recently investigated mechanism leading to VML-induced force loss [15]. Recent investigations support an increased proportion of type I fibers following VML, and overall contractile slowing of the muscle [15,53]. Herein, a greater proportion of type I fibers and a smaller proportion of type IIx and IIb fibers were present following VML. Supporting that type I fibers have a greater number of mitochondria and display a preference for fatty acids as substrates for ATP production [54], alteration in fiber type proportions corresponded with increased lipid and decreased carbohydrate oxidation. Greater type I fibers also mirrors the increased numbers of fibers staining for NADH, further indicating slowing of the muscle phenotype. Fast muscle fibers, particularly type IIb, are the first to degenerate in models of DMD, resulting in a higher proportion of type I fibers, contributing in part to decreases in force [55]. Activation of skeletal muscle fibers occurs through recruitment of motor units in order of size, from smaller, slow-twitch fatigue resistant motor units to larger, fast-fatigable motor units; which can be identified with $MyHC_{slow}$ and $MyHC_{2B}$ isoform expression, respectively indicating type I and IIb fibers [56,57]. Orderly motor unit recruitment occurs, in part based on the magnitude of force generation required. The greater proportion of type I fibers following VML suggests that fewer high force motor units may be recruited, leading to decreased maximal force generation and contributing to chronic force loss. The altered fiber type proportions may signify preferential preservation of low force functions, such as ambulation, which require predominantly slow-twitch motor units. This sacrifices the ability to generate high-force contractions to protect the muscle during the wound healing and repair process.

Collectively, the muscle-specific and whole-body metabolic consequences and chronic loss of function following VML may explain the limited regenerative response to rehabilitation in

the muscle remaining after VML. Lack of physiologic responses to exercise and rehabilitation is clear both preclinically [16,17,30,58] and clinically [59,60], but the mechanisms limiting these responses are unclear. Typically, metabolic flexibility improves in response to exercise training or rehabilitation, with increases in RER during exercise expected due to increased metabolic demand. Following VML, underlying pathophysiology and a combination of factors within the muscular environment, including mitochondrial dysfunction, altered fiber type proportions, and disproportionate capillary per fiber distribution, may in part explain the blunted muscle response. Additionally, heightened and prolonged inflammatory responses following VML can drive excessive extracellular matrix deposition, overexpression of pro-fibrotic factors, such as TGF-β1 and CCN2/CTGF, and pathologic fibrosis [40,42]. This is a similar response to that seen in the skeletal muscle of Duchenne muscular dystrophy-afflicted patients and animal models, contributing in part to impairment of the muscle's regenerative and oxidative capacity [61].

Clinically, long periods of physical inactivity are detrimental to skeletal muscle. It is expected that, following VML, patients will have limited physical activity during recovery and repair periods. However, contrary to our hypothesis, there was no VML-induced change in physical activity identified here in a model of VML injury. Despite similar levels of physical activity following VML injury, metabolic flexibility and metabolic rate flux were dysregulated. Future investigations should model VML concurrent with inactivity [62], such as restricting cages [63,64], to provide greater insight into the interplay of physical and metabolic activity following VML.

## Acknowledgments

Work was completed using the TissueScope LE slide scanner and the C2 Nikon Confocal microscope at the University of Minnesota—University Imaging Centers.

## Author Contributions

**Conceptualization:** Christiana J. Raymond-Pope, Jarrod A. Call, Sarah M. Greising.

**Data curation:** Kyle A. Dalske, Christiana J. Raymond-Pope, Jennifer McFaline-Figueroa, Alec M. Basten.

**Formal analysis:** Kyle A. Dalske, Christiana J. Raymond-Pope, Jennifer McFaline-Figueroa, Alec M. Basten, Jarrod A. Call, Sarah M. Greising.

**Funding acquisition:** Jarrod A. Call, Sarah M. Greising.

**Investigation:** Kyle A. Dalske, Christiana J. Raymond-Pope, Alec M. Basten.

**Methodology:** Kyle A. Dalske.

**Project administration:** Christiana J. Raymond-Pope, Jarrod A. Call, Sarah M. Greising.

**Supervision:** Jarrod A. Call.

**Writing – original draft:** Kyle A. Dalske, Christiana J. Raymond-Pope, Jarrod A. Call, Sarah M. Greising.

**Writing – review & editing:** Christiana J. Raymond-Pope, Jennifer McFaline-Figueroa, Alec M. Basten, Jarrod A. Call, Sarah M. Greising.

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
