## [Decision Letter · Decision Letter 0]

14 May 2021

PONE-D-21-12451

Independent of physical activity, volumetric muscle loss injury impairs whole-body metabolism

PLOS ONE

Dear Dr. Greising,

Thank you for submitting your manuscript to PLOS ONE. After careful consideration, we feel that it has merit but does not fully meet PLOS ONE’s publication criteria as it currently stands. Therefore, we invite you to submit a revised version of the manuscript that addresses the points raised during the review process.

The reviewers have raised some valid concerns especially about how to quantification of data. Please address those issues.   

We look forward to receiving your revised manuscript.

Kind regards,

Ashok Kumar

Academic Editor

PLOS ONE

Journal Requirements:

2. Please specify the animal model used in the title of your article.

3.Your ethics statement should only appear in the Methods section of your manuscript. If your ethics statement is written in any section besides the Methods, please delete it from any other section.

Reviewers' comments:

Reviewer's Responses to Questions

**Comments to the Author**

1. Is the manuscript technically sound, and do the data support the conclusions?

Reviewer #1: Yes

Reviewer #2: Partly

2. Has the statistical analysis been performed appropriately and rigorously? 

Reviewer #1: Yes

Reviewer #2: Yes

3. Have the authors made all data underlying the findings in their manuscript fully available?

Reviewer #1: Yes

Reviewer #2: Yes

4. Is the manuscript presented in an intelligible fashion and written in standard English?

Reviewer #1: Yes

Reviewer #2: Yes

5. Review Comments to the Author

Reviewer #1: In the manuscript entitled ‘Independent of physical activity, volumetric muscle loss injury impairs whole-body metabolism’ Dlaske et al performed a longitudinal study exploring the effects of volumetric muscle loss (VML) on whole body physical activity and metabolism. In adult C57BL/6J mice, they determined that although VML injury did not affect physical activity measured in terms of ambulatory distances but adversely affected metabolism. Post 6 weeks of VML injury, there was significant reduction in 24 h metabolic rate, and 24 h RER. Estimation of muscle function showed a significant decrease in maximal isometric torque post 8 weeks of VML injury. At biochemical level, the authors found the levels of citrate synthase, complex 1 and SDH activity to be comparable between control and VML injured GA tissue. Authors did histological studies to determine signs of fibrosis and more centrally located nuclei overall 8 weeks post VML. Interestingly, there was significantly more NADH positive myofibers post 4 weeks of VML injury and the numbers of NADH positive myofibers were comparable between control and post 8 weeks of VML injury. Furthermore, the authors report prevalence of more myofibers with smaller CSA in VML injured GA compared to control. Also, VML groups had more Type I and Type IIA fibers at 4 weeks.

Overall, the manuscript is well-written with detailed analysis of experimental data. The report provides new insight into fibertype distribution following VML which may attribute to the muscle specific metabolic alterations.

Comments

1. The bar diagrams throughout the manuscript are inconsistent with regard to style. Authors should have similar design for all bar diagrams in the figures.

2. In Figure 3B, authors should indicate individual data points.

3. The last panel of figure 4B showing fiber type specific differences should be integrated in Figure 6.

Reviewer #2: Dalske et al. investigated the effect of volumetric muscle loss (VML) on physical activity and metabolism using male C57BL6 mice. They monitored cage activity and O2/CO2 consumption for RER measurement in the chamber, measured muscle function and performed enzymatic activity assay and histological analysis using gastrocnemius muscles. VML induced the loss of muscle function and decreased metabolic rate and RER associated with increased carbohydrate oxidation and reduced lipid oxidation without changing the physical activity levels. While no enzyme activity was altered in the muscle mitochondria, VLM caused fiber type change toward the increased ratio of oxidative fibers with minimal changes in capillary density mitochondrial enzyme expressions. While this manuscript potentially gives us new insight to understand VML injury clinically, the data shown here support their conclusion partially. In particular, it is hard to judge that VML changed whole-body metabolism. Furthermore, based on my understanding, few novel data is presented in this manuscript that limits enthusiasm for this manuscript.

Major concerns:

1. When discussing the change of whole-body metabolism, more data from the other organs (e.g., fat and liver) are essential.

2. In general, histological analysis is qualitative, not quantitative, and thus not a reliable method to detect subtle changes in protein expressions. Please consider using another way to quantify them.

3. Considering metabolic inflexibility, nutrient overload, and heightened substrate competition are hypothetical events in obese patients. Thus, showing food intake and the levels of glucose and fatty acid in the circulation would be necessary.

4. In discussion, the authors speculate the hypoxia based on decreased capillary distribution. In my opinion, increased HIF-a expression elevates GLUT1 expression in the muscle, causing a switch from lipid oxidation to glycolysis due to a lower level of oxygen available (ref. 44). Enzyme activity data show no change occurred in the muscle, and histological data describes the minimal changes of the enzyme expressions. If Fig2 E&F are true, how did it happen? Please explain.

Minor concerns:

1. Line 302. It says, “At baseline the 24-hour RER corresponded to a carbohydrate oxidation of 6.42±1.17 g/min.” But Fig 1C shows the level of 24-hr carbohydrate oxidation appears to be less than 6.0. Which is correct?

2. Line 324. What is delta metabolic rate? Please explain.

3. Line 329. While AUC is discussed, no AUC is shown in Fig 2B, which is confusing.

4. Line 361. Same as above. No AUC data is presented in Fig 2C, despite referring to the figure in the sentence.

5. Line 372. Table 1 and Fig 3 are redundant. I don’t think Fig 3 is necessary. Please consider.

6. Line 424. Different than control -> Different from control?

6. PLOS authors have the option to publish the peer review history of their article (what does this mean?). If published, this will include your full peer review and any attached files.

Reviewer #1: No

Reviewer #2: No

---

## [Author Response · Author response to Decision Letter 0]

9 Jun 2021

Response to Comments by Editor

Comment 1: Please ensure that your manuscript meets PLOS ONE's style requirements, including those for file naming.

Response: We confirm all formats and names are correct. 

Comment 2: Please specify the animal model used in the title of your article.

Response: The article title has been modified to specify the animal model used; Independent of physical activity, volumetric muscle loss injury in a murine model impairs whole-body metabolism.

Comment 3: Your ethics statement should only appear in the Methods section of your manuscript. If your ethics statement is written in any section besides the Methods, please delete it from any other section.

Response: Our ethics statement now only appears in the Methods section of the manuscript. 

Response to Comments by Reviewer 1

Comment: In the manuscript entitled ‘Independent of physical activity, volumetric muscle loss injury impairs whole-body metabolism’ Dlaske et al performed a longitudinal study exploring the effects of volumetric muscle loss (VML) on whole body physical activity and metabolism. In adult C57BL/6J mice, they determined that although VML injury did not affect physical activity measured in terms of ambulatory distances but adversely affected metabolism. Post 6 weeks of VML injury, there was significant reduction in 24 h metabolic rate, and 24 h RER. Estimation of muscle function showed a significant decrease in maximal isometric torque post 8 weeks of VML injury. At biochemical level, the authors found the levels of citrate synthase, complex 1 and SDH activity to be comparable between control and VML injured GA tissue. Authors did histological studies to determine signs of fibrosis and more centrally located nuclei overall 8 weeks post VML. Interestingly, there was significantly more NADH positive myofibers post 4 weeks of VML injury and the numbers of NADH positive myofibers were comparable between control and post 8 weeks of VML injury. Furthermore, the authors report prevalence of more myofibers with smaller CSA in VML injured GA compared to control. Also, VML groups had more Type I and Type IIA fibers at 4 weeks.

Overall, the manuscript is well-written with detailed analysis of experimental data. The report provides new insight into fiber type distribution following VML which may attribute to the muscle specific metabolic alterations.

Response: We thank the reviewer for their time and for providing a concise summary of the study and associated data presented therein. We have provided responses to each comment posited below.

Comment 1: The bar diagrams throughout the manuscript are inconsistent with regard to style. Authors should have similar design for all bar diagrams in the figures.

Response: Thank you for your comment. Across all figures, bar diagrams have been formatted for consistency, changes were made to Figures 4 and 5.

Comment 2: In Figure 3B, authors should indicate individual data points.

Response: Thank you for your suggestion. In response to this comment and Reviewer 2’s comment, we have deleted Figure 3 as the data in this figure is also presented in Table 1.

Comment 3: The last panel of figure 4B showing fiber type specific differences should be integrated in Figure 6.

Response: Thank you for your suggestion. We moved the final panel with the MyHC staining to the new Figure 5.

Response to Comments by Reviewer 2

Comment: Dalske et al. investigated the effect of volumetric muscle loss (VML) on physical activity and metabolism using male C57BL6 mice. They monitored cage activity and O2/CO2 consumption for RER measurement in the chamber, measured muscle function and performed enzymatic activity assay and histological analysis using gastrocnemius muscles. VML induced the loss of muscle function and decreased metabolic rate and RER associated with increased carbohydrate oxidation and reduced lipid oxidation without changing the physical activity levels. While no enzyme activity was altered in the muscle mitochondria, VLM caused fiber type change toward the increased ratio of oxidative fibers with minimal changes in capillary density mitochondrial enzyme expressions. While this manuscript potentially gives us new insight to understand VML injury clinically, the data shown here support their conclusion partially. In particular, it is hard to judge that VML changed whole-body metabolism. Furthermore, based on my understanding, few novel data is presented in this manuscript that limits enthusiasm for this manuscript.

Response: We thank the reviewer for their time and the assessment of our work. We have responded to each comment below.

Major concerns:

Comment 1: When discussing the change of whole-body metabolism, more data from the other organs (e.g., fat and liver) are essential.

Response: This is an important point and will be considered in future investigations. In the current study, our primary interest is skeletal muscle tissue. Other tissue types are outside of the scope of our aims, and we did not harvest additional tissues for these analyses. In the future, we could design additional studies to investigate other tissue types following VML injury.

Comment 2: In general, histological analysis is qualitative, not quantitative, and thus not a reliable method to detect subtle changes in protein expressions. Please consider using another way to quantify them.

Response: Thank you for your comment. In addition to histological analysis of NADH and SDH activity, corresponding to Complex I and II of the electron transport chain, respectively, we did biochemically examine enzyme kinetics. Although western blot assessment of MyHC isoforms was considered, quantification of fiber type expression across regions of interest is a common evaluation in the literature and a precise method for fiber type quantification. Additionally, our fiber type observations are similar to findings in recent studies of VML, with an increased proportion of type I fibers and decreased proportion of type II fibers following injury. Further, investigators were blinded during the staining, imaging, and post-imaging analyses for all histology, ensuring the quality of evaluations reported herein. The three regions of interest were standardized across muscles. The blinded nature of assessments in this study has been made clearer in the methods section.

Comment 3: Considering metabolic inflexibility, nutrient overload, and heightened substrate competition are hypothetical events in obese patients. Thus, showing food intake and the levels of glucose and fatty acid in the circulation would be necessary.

Response: Thank you for this question. Although we did not directly measure food intake, we did record weekly body mass for all mice. Because body mass was not different over time, we do not expect there to be differences in food intake. The lack of difference for mass was noted in the text and shown here graphically for clarity. We do not have any additional data on circulating glucose or fatty acid oxidation to add to this work.

Comment 4: In discussion, the authors speculate the hypoxia based on decreased capillary distribution. In my opinion, increased HIF-a expression elevates GLUT1 expression in the muscle, causing a switch from lipid oxidation to glycolysis due to a lower level of oxygen available (ref. 44). Enzyme activity data show no change occurred in the muscle, and histological data describes the minimal changes of the enzyme expressions. If Fig2 E&F are true, how did it happen? Please explain.

Response: Thank you for your comment. Within the discussion we provide a proposed explanation for increased lipid oxidation following VML injury, as shown in Fig 2. Similar observations of increased lipid oxidation have been reported in the literature in pathological conditions, including in cardiac tissue of animals and humans with heart failure, diabetes, and during reperfusion following ischemia, as well as in models of amyotrophic lateral sclerosis. The increased lipid oxidation may partially explain the increased levels of NADH observed herein, as each turn of beta-oxidation produces one molecule of NADH. Therefore, a 16-carbon fatty acid, for example, produces 7 molecules of NADH, greater than the combined number of NADH molecules produced during glycolysis and the subsequent conversion of pyruvate to acetyl-coA. The preceding discussion has been added to the expanded Discussion section of the manuscript.

Minor concerns:

Comment 5: Line 302. It says, “At baseline the 24-hour RER corresponded to a carbohydrate oxidation of 6.42±1.17 g/min.” But Fig 1C shows the level of 24-hr carbohydrate oxidation appears to be less than 6.0. Which is correct?

Response: Thank you for pointing out this error. Fig 1C displays the correct mean and standard deviation for carbohydrate oxidation. The text within the Results section has been changed accordingly.

Comment 6: Line 324. What is delta metabolic rate? Please explain.

Response: Delta metabolic rate is the difference between the metabolic rate during the active and inactive phases of the light cycle. Delta metabolic rate reflects the metabolic rate flux, as described in Lines 135-137 and 279-280. We have made this clearer on Line 324.

Comment 7: Line 329. While AUC is discussed, no AUC is shown in Fig 2B, which is confusing.

Response: AUC data for the metabolic rate specifically is shown in Fig 2B.

Comment 9: Line 361. Same as above. No AUC data is presented in Fig 2C, despite referring to the figure in the sentence.

Response: Thank you for your comment. The correct reference for the RER AUC data is Fig 2D. This reference has been changed accordingly.

Comment 10: Line 372. Table 1 and Fig 3 are redundant. I don’t think Fig 3 is necessary. Please consider. 

Response: Thank you for your suggestion. We deleted this figure in response to this comment and the comment stated by Reviewer 1. 

Comment 11: Line 424. Different than control -> Different from control?

Response: We have changed the wording to indicate different from control.

---

## [Editor Report · Decision Letter 1]

10 Jun 2021

Independent of physical activity, volumetric muscle loss injury in a murine model impairs whole-body metabolism

PONE-D-21-12451R1

Dear Dr. Greising,

We’re pleased to inform you that your manuscript has been judged scientifically suitable for publication and will be formally accepted for publication once it meets all outstanding technical requirements.

Kind regards,

Ashok Kumar

Academic Editor

PLOS ONE
---

## [Editor Report · Acceptance letter]

17 Jun 2021

PONE-D-21-12451R1 

Independent of physical activity, volumetric muscle loss injury in a murine model impairs whole-body metabolism 

Dear Dr. Greising:

I'm pleased to inform you that your manuscript has been deemed suitable for publication in PLOS ONE. Congratulations! Your manuscript is now with our production department. 

Kind regards, 

on behalf of

Dr. Ashok Kumar 

Academic Editor

PLOS ONE